**Data Availability Statement:** The data underlying the results presented in the study are available from the University of Melbourne (https://melbourneinstitute.unimelb.edu.au/hilda).

# Resilience of self-reported life satisfaction: A case study of who conforms to set-point theory in Australia

**Ida Kubiszewski**[1]*, **Nabeeh Zakariyya**[2], **Robert Costanza**[1], **Diane Jarvis**[3]

**1** Crawford School of Public Policy, The Australian National University, Canberra, ACT, Australia, **2** Research School of Economics, The Australian National University, Canberra, ACT, Australia, **3** College of Business, Law & Governance, James Cook University, Douglas, QLD, Australia

* ida.kub@gmail.com

## Abstract

While self-reported life satisfaction (LS) has become an important research and policy tool, much debate still surrounds the question of what causes LS to change in certain individuals, while not in others. Set-point theory argues that individuals have a relatively resilient LS or "set point" (i.e. there is a certain LS level that individuals return to even after major life events). Here, we describe the extent to which LS varies over time for 12,643 individuals living in Australia who participated in at least eight annual waves of the Household Income and Labour Dynamics in Australia (HILDA) Survey. We use the standard deviation (SD) of year-on-year LS by individuals (SD of LS) as a measure of instability and an inverse proxy for resilience. We then model SD of LS as the dependent variable against average LS scores over time by individual, Big Five personality scores by individual, the number of waves the individual participated in, and other control variables. We found that SD of LS was higher (lower resilience) in participants with a lower average LS and greater degrees of extraversion and agreeableness. Set-point theory thus applies more to individuals whose average LS is already high and whose personality traits facilitate higher resilience. We were able to explain about 35% of the stability in LS. These results are critical in designing policies aimed at improving people's lives.

## Introduction

What contributes to human wellbeing, and to what extent, has been a growing debate within political and academic spheres over the past decades [1, 2]. Biological, psychological, ecological, and societal factors all contribute to human wellbeing in complex ways over multiple temporal and spatial scales. However, most studies around human wellbeing approach these factors disjointedly, from different fields.

What is known, is that increased subjective wellbeing is related to many aspects of life [3], including marriage [4], friendship [5], overall social support network [6], income [7], work performance [8], mental [9] and physical [10] health, and even longevity [11]. Beyond the individual, the costs of low levels of subjective wellbeing can also be directly seen within society.

**Funding:** This research was partially funded by the Australian Government through the Australian Research Council on a Discovery Early Career Researcher Award (ProjectID: DE150100494)."There was no additional external funding received for this study.

**Competing interests:** The authors have declared that no competing interests exist.

For example, individuals that report to be suffering can cost a company three times more in sick leave than individuals that report to be thriving [12]. This does not take into account the loss of productivity while these individuals are at work. Poor mental health has been shown to reduce gross domestic product (GDP) by up to 5% in OECD countries [13]. More indirect costs also exist. Inequality in subjective wellbeing erodes trust within a society, a crucial aspect of societal cohesion [14]. As more individuals report lower subjective wellbeing, social relationships, trust levels, and a sense of belonging in a community suffer [15–17].

Research has shown that factors that are characteristics of individuals, like personality, explain approximately 30–40% of subjective wellbeing, while 60–70% can be attributed to environmental factors [18–22]. Researchers have been attempting to understand these factors influencing subjective wellbeing in populations for decades, especially whether subjective wellbeing can be changed over the longer-term [20]. Set-point theory states that individuals' wellbeing is relatively stable and resilient over the long-term. It asserts that distinct events impact an adult individual's subjective wellbeing positively or negatively but only temporarily, eventually returning to a stable baseline or set-point [23, 24]. Thus, according to this theory, each individual has a unique set-point, to which they return. Set-point theory has dominated the field for over 30 years [25], however, in recent years it has been criticized as certain major life events have been shown to cause permanent changes to subjective wellbeing [26–33].

There are moderating factors that can begin to explain these contrasting findings around set-point theory. One such factor is an individual's personality traits, which are defined as relatively stable, cross-situational patterns of thought, feeling, motivation, and behaviour [34]. In this paper, we use the Big Five personality dimensions (openness, conscientiousness, extraversion, agreeableness, and neuroticism or its inverse emotional stability), which have all been shown to be associated in some way with indicators of subjective wellbeing [35, 36]. What remains to be investigated is whether these personality traits influence the stability and resilience in subjective wellbeing over time.

The related concepts of variation, stability, resilience, and set-point are used in several fields with a range of definitions. In ecology, resilience is often defined as the capacity of a system to adapt and change in response to shocks in a manner that retains its fundamental identity [37]. A related concept has been applied to wellbeing, but usually labelled as stability [25, 38, 39]. Stability is a system's ability to remain unchanged after a shock. Although similar, stability and resilience are not identical. An increased stability within a system does not always lead to greater resilience. A system may need a bit of instability to increase its resilience, and help fortify the system against future shocks by allowing it to adapt and change. For example, the resilience of a coral reef is demonstrated by its capacity to absorb disturbances and reorganize, while undergoing some limited change, retaining essentially the same function, structure, identity, and feedbacks [40]. If the reef loses resilience it can transform to a much less desirable macroalgae based system, which is itself very resilient and difficult to recover from (Fig 1). Likewise social resilience describes the capacity of societies and individuals to cope and adapt to change.

The concept of resilience can be applied when looking at human wellbeing, or more narrowly, self-reported life satisfaction (LS). Humans experience shocks through both day-to-day experiences and major life events. A good analogy for LS resilience is the operation of an individual's immune system at the biological level. Individuals with well-functioning immune systems can both resist infections and bounce back from them more easily than individuals with compromised immune systems. A resilient individual has a well-functioning immune system and we would expect to see less variation over time in their health status.

Likewise, how individuals' LS responds to shocks, and whether they return to their set-point, provides us with critical information about an individual's resilience and ability to live a

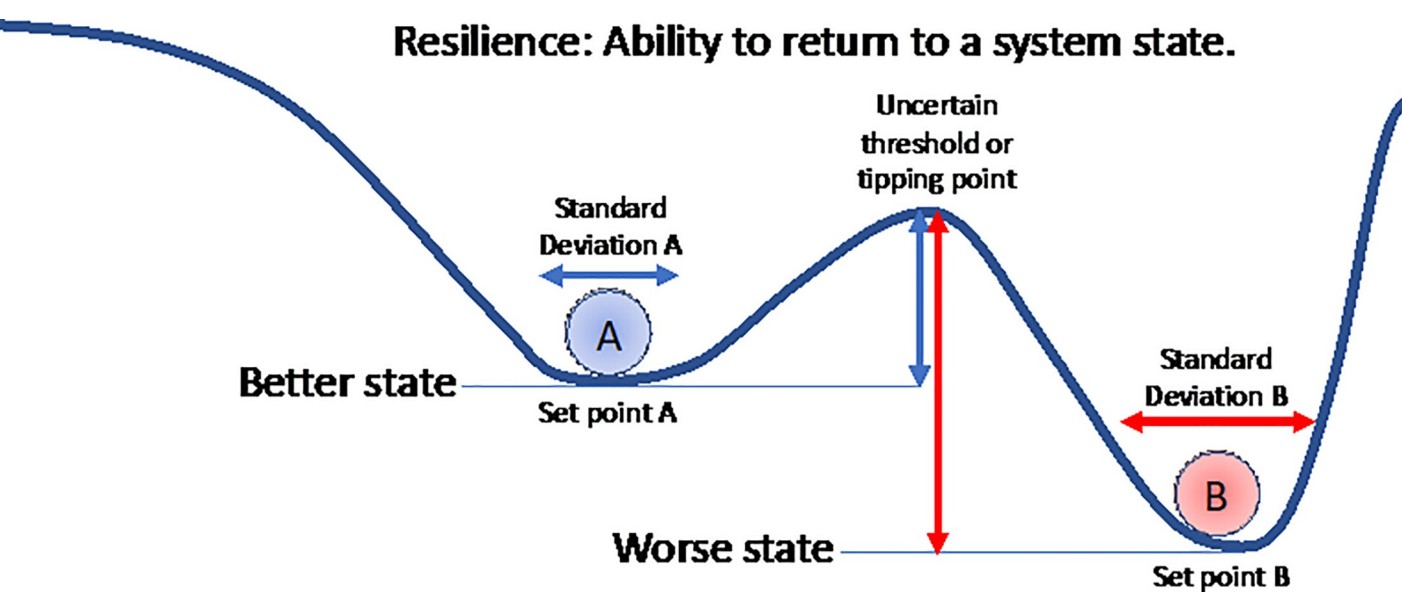

**Fig 1. Resilience is the size of the vertical arrows in this diagram–how far can the system (individual, ecosystem, economy, etc.) be perturbed and still return to its original state (or "set-point").** For ball A, the arrow is a measure of how far the system can be perturbed before falling to a worse state. For ball B, it is a measure of how much resilience must be overcome to enable transition to a better state. High resilience is positive if it prevents transition to worse states but not if it prevents transition to better states.

productive life. However, resilience is hard to measure directly, especially before the fact. Using the immune system analogy, it is hard to know how good someone's immune system is until it has been challenged, and even then, we do not know how big a challenge it could resist.

In this study, we use the standard deviation of life satisfaction (SD of LS) for an individual over time as a measure of the variation in their LS. We assume that the smaller this variation is, the higher their resilience. Resilience is the ability to maintain an equilibrium [41], as summarized in Fig 1. While the standard deviation or variation over time is not a direct measure of resilience, we can assume that the smaller the SD the less likely the system will reach a threshold or tipping point and thus has both higher stability and higher resilience. We can see that resilience is difficult to measure directly because the location of the threshold or tipping point is usually uncertain or unknown. In addition, it is important to recognize that resilience is just one characteristic of a system. It is a neutral term, neither positive or negative. For example, high resilience may prevent transformation to a better state, as in recovery from a macroalgae based reef to a coral reef, in which case it is something to be overcome not enhanced.

Many reports [42, 43] that analyse societal contributions to wellbeing focus on national averages. Although important for international comparisons and understanding the overall wellbeing of a nation, these indicators and reports: (1) omit the distribution of wellbeing within a country; (2) do not look at how individual LS changes over time; and (3) fail to identify and study those most at risk [44, 45].

The goal of this paper is to understand how self-reported LS changes over time and how its resilience or stability varies as a function of a range of variables, including average level of LS and different aspects of personality.

In this paper, we focus on self-reported LS, a component of wellbeing, and its stability and resilience over time. We analyse the resilience of LS of different portions of the Australian population by using the standard deviation on life satisfaction (SD of LS) as a proxy for resilience, and in turn, the ability of an individual to return to their set-point. We look at how average LS over time and personality scores affect individual's resilience of LS.

## Methods and data

### Sample selection

In this paper, we use data from waves 1–17 of the Household and Income Labour Dynamics in Australia (HILDA) Survey. These waves correspond to years 2001–2017. After removing individuals that did not respond to our main variable (overall LS), there are 31,194 individuals in the HILDA Survey sample.

Using a strongly balanced panel of only participants that participated in all 17 waves reduces the sample size to 28% of the full sample. Thus, we restrict our sample to those individuals who responded in at least 8 waves out of 17 with a maximum gap of 3 years. This accounts for around 41% of the full sample, or 12,643 individuals. For robustness, we compare our main sample with a strongly balanced panel of those who responded in all 17 waves.

### Main variables and metrics

In each wave of the HILDA Survey, respondents are asked "All things considered, how satisfied are you with your life?" Responses are given on an 11-point Likert scale where 0 corresponds to totally dissatisfied and 10 stands for totally satisfied. We acknowledge that calculating the average of Likert items can be problematic, especially not knowing whether increments in scale correspond to equal increments in the underlying latent variable. Treating life satisfaction (LS) as ordinal versus interpersonally cardinally comparable is a contentious issue in the literature. Justifications for cardinality shows that treating LS data as cardinal yields similar results to treating it as ordinal, and both assumptions are compatible with LS scores [46–48]. Further, Kristoffersen shows that LS scores are equidistant [48]. The purpose of this paper does not require us to take a strong stand in this debate.

For each individual in our sample, we examine the distribution of their LS scores across all the years they responded to the LS question. We summarize this distribution over time by constructing first and second order moments.

The first order moments gives the average LS of an individual over time

$$\overline{LS}_i = \frac{\sum_{t=1}^{T} LS_{it}}{T_i} \qquad (1)$$

where $LS_{it}$ is the LS score for individual $i$ in year $t$ ('average LS').

$T_i$ is the total number of years the individual responded to the question.

The second order moment gives the variations of an individual's LS across years. To do this, we use the standard deviation of LS for each individual over time. This is given by

$$\sigma_i^{LS} = \sqrt{\frac{\sum_{t=1}^{T} (LS_{it} - \overline{LS}_i)}{T_i}} \qquad (2)$$

We use $\sigma_i^{LS}$ as the main metric to measure resilience in an individual's LS. An individual with $\sigma_i^{LS} = 0$ reported no change in their LS score during this period. A higher $\sigma_i^{LS}$ implies higher variability (and lower resilience to life events). $\sigma_i^{LS}$ is agnostic to the level of average LS of the individual. That is, whether an individual has a low average LS or a high average LS over the years they responded to the Survey, $\sigma_i^{LS}$ measures the deviation from that respective average. This allows us to compare variability between individuals even if they report different levels of average LS.

We divide individuals based on their average LS in to three separate groups: (*i*) suffering (0–4), (*ii*) struggling (5–6), and (*iii*) thriving (7–10). Suffering includes those with an average

**Table 1.** *Distribution of people's life satisfaction.*

|  | Suffering (0–4) | Struggling (5–6) | Thriving (7–10) |
|---|---|---|---|
| **Denmark** | 3.0% | 5.1% | 91.9% |
| **Finland** | 3.6% | 7.9% | 88.5% |
| **Iceland** | 4.1 | 8.5 | 87.3 |
| **United Kingdom** | 9.6% | 15.5% | 74.9% |
| **France** | 17.0% | 23.4% | 59.6% |
| **Russia** | 26.9% | 34.7% | 38.4% |

This is a modified version of Table 2 from [50].

LS of 4.4 or less. Struggling includes those with an average LS between 4.5 and 6.4. Thriving includes those with an average LS of 6.5 or greater. These three groupings are derived from a Gallup groupings [49] and utilized in an analysis of European county LS [50]. Table 1 provides the results this LS study in certain European countries as a means of comparison. Some of these differences may be due to different cultural perceptions [51, 52].

## Correlates of LS variability

Our analysis focuses on answering two key questions: (*i*) is there a correlation between $\sigma_i^{LS}$ (standard deviation of LS) and average LS by individual? And, (*ii*) is there a correlation between $\sigma_i^{LS}$ and the different personality traits by individual?

In order to answer question (*ii*), we use the Big Five personality traits that are derived by the HILDA Survey using a 36-item inventory [53]. These traits are extroversion, agreeableness, conscientiousness, emotional stability, and openness to experience. Between waves 1–17, personality questions are only available in waves 5, 9, 13 and 17. The questions used in the HILDA Survey are a short version of Saucier's [54] 'Big Five' personality test. For each individual, we obtain their average and standard deviation for each personality trait. The formulae are similar to those of average LS and $\sigma_i^{LS}$ where we average over the years that an individual responded to the personality questions.

We examine raw correlations between $\sigma_i^{LS}$ and these variables as well as through ordinary least squares regressions. The regression model is given by

$$\sigma_i^{LS} = \alpha + \beta \ \overline{LS}_i + \gamma \underline{P}_i + \rho \underline{X}_i + u_i \tag{3}$$

where $\underline{P}_i$ is a vector of personality scores and $\underline{X}_i$ is a vector of control variables. In our unbalanced sample, we control for the number of years an individual responded. We also include control variables found to be important in previous LS research, such as average household income (generally positive relationship with LS) and age (generally having a positive or U shaped relationship with LS).

## Results

Approximately 91% of the 12,643 respondents that participated in at least 8 waves (referred to as our unbalanced sample), reported an average life satisfaction (LS) between 6.5 and 10. We grouped these individuals into the 'thriving' group. Approximately 8% of the participants reported that they are 'struggling' (average 4.5–6.4 on the LS scale) and 1% are 'suffering' (0–4.4 on the LS scale) (Fig 2). These groups are loosely based on a Gallup grouping scheme [49] Table 2 provides summary statistics for our unbalanced sample in the three groups. Supplementary S1 Table provides the same statistics but for our balanced group (where respondents

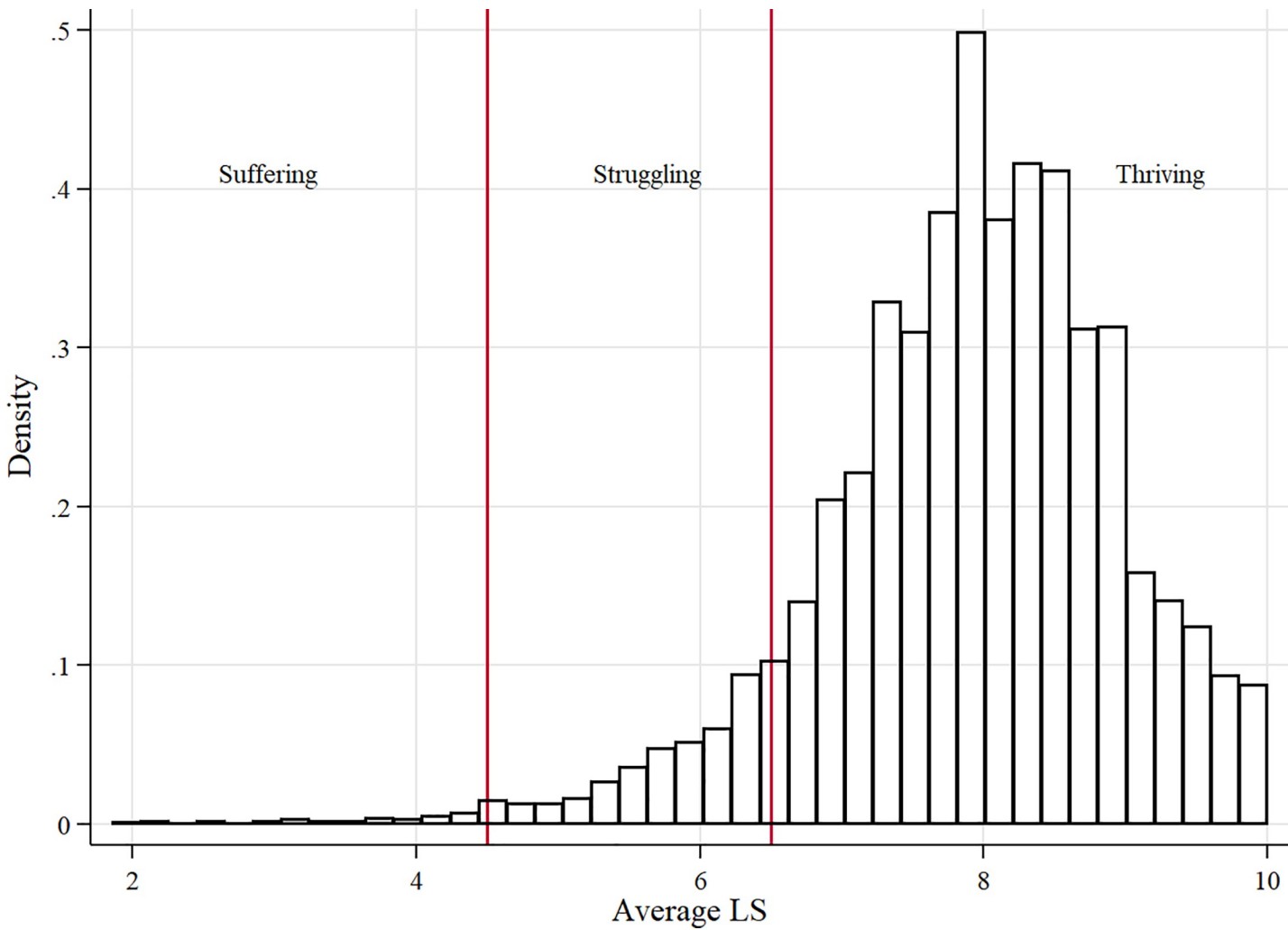

**Fig 2. Distribution of average LS over time divided into suffering (LS = 0–4.4), struggling (LS = 4.5–6.4) and thriving (LS = 6.5–10).**

participated in all 17 waves). For each sub-sample, and the whole sample, the Table 2 shows the number of respondents (N), as well as the averages across individuals of: the average LS within an individual over time, the average personality scores within an individual over time, the average household disposable income over time, and age. It also shows the standard deviation of each of those averages.

We also graphed the SD of LS for our unbalanced sample and found a clear distinction between the three groups (Fig 3). Individuals in the suffering range had an average SD of 1.96, those in the struggling range an average of 1.54, and those that were thriving an average SD of 0.87. Fig 4 is a plot of the LS and SD of LS scores for all individuals in each of our three groups, showing a clear trend between the three groups.

Just as we find a trend in the relationship between SD of LS and LS *across* the three groups, we also see trends between SD of LS and the averages of extraversion, conscientiousness, emotional stability, openness, and household disposable income *within* the 3 groups (suffering, struggling and thriving). We also see trends in the SD *within* all the groups (Table 2, column 1–3). We also see trends in the standard deviation within individuals around agreeableness, conscientiousness, openness, household disposable income, and age. Table 2, column 5–7

**Table 2. Summary statistics, unbalanced sample.**

| | Within person averages over time *(SD between averages of LS of individuals)* | | | | Within person SDs over time *(SD between SD of individuals)* | | | |
|---|---|---|---|---|---|---|---|---|
| | Suffering (1) | Struggling (2) | Thriving (3) | All (4) | Suffering (5) | Struggling (6) | Thriving (7) | All (8) |
| N | 81 | 1,102 | 11,550 | 12,643 | 81 | 1,102 | 11,550 | 12,643 |
| Life satisfaction | 3.55 | 5.84 | 8.12 | 7.91 | 0.71 | 0.53 | 0.79 | 1.05 |
| | (1.96) | (1.54) | (0.87) | (0.93) | (0.66) | (0.58) | (0.42) | (0.48) |
| Extroversion | 3.82 | 4.05 | 4.46 | 4.42 | 1.05 | 0.94 | 0.97 | 0.98 |
| | (0.55) | (0.52) | (0.48) | (0.48) | (0.35) | (0.33) | (0.30) | (0.30) |
| Agreeableness | 5.21 | 5.15 | 5.41 | 5.39 | 1.02 | 0.90 | 0.80 | 0.81 |
| | (0.61) | (0.55) | (0.49) | (0.50) | (0.48) | (0.39) | (0.36) | (0.36) |
| Conscientiousness | 4.53 | 4.75 | 5.12 | 5.09 | 1.13 | 0.95 | 0.90 | 0.92 |
| | (0.62) | (0.53) | (0.50) | (0.50) | (0.41) | (0.35) | (0.33) | (0.33) |
| Emotional stability | 4.28 | 4.74 | 5.27 | 5.22 | 1.05 | 0.92 | 0.93 | 0.94 |
| | (0.72) | (0.63) | (0.56) | (0.57) | (0.64) | (0.39) | (0.37) | (0.37) |
| Openness | 4.38 | 4.28 | 4.18 | 4.19 | 1.18 | 1.02 | 0.96 | 0.97 |
| | (0.57) | (0.54) | (0.51) | (0.51) | (0.48) | (0.37) | (0.33) | (0.33) |
| HH disposable income | 50,778 | 62,368 | 83,718 | 81,798 | 30,450 | 36,667 | 54,190 | 53,252 |
| | (27,344) | (32,365) | (38,376) | (37,824) | (32,489) | (37,183) | (51,590) | (50,518) |
| Age | 48.51 | 45.00 | 44.76 | 44.80 | 12.86 | 15.56 | 18.32 | 18.08 |
| | (4.57) | (4.45) | (4.40) | (4.41) | (0.81) | (0.88) | (0.90) | (0.89) |

This table summarizes the primary variables for our 3 subsamples and the whole sample. Columns 1–4 provide the average across individuals ($\overline{LS}_i$) of the average LS within an individual over time ($LS_{it}$) [see Eq 1]. The (parenthesis) provide the standard deviation between the average LS across individuals. Columns 5–8 provide the average across individuals of the standard deviations within each individual over time ($\sigma_i^{LS}$) [see Eq 2]. The (parenthesis) provide the standard deviation among the standard deviations of each individual over time.

show the average across individuals of the SD within an individual over time (SD between SD of individuals).

We also ran an OLS regression model to investigate the relationship between the SD of LS and the average of LS for individuals that responded to at least 8 waves, controlling for the number of waves they participated in. We found a strong negative correlation ($R^2 = 0.233$) (Table 3, column 1). This shows that as an individual's average LS increases, their SD of LS decreases (their resilience increases).

We also ran multiple regression models with SD of LS within each individual over time as the dependent variable and a series of independent variables. As more variables were included into the regression, including the Big Five personality types, the standard deviation (SD) of each of the personality types, income, SD of income, age, and number of waves an individual answered, we found that the regression became stronger, as is to be expected. Most of the variables were significant, except for conscientiousness which became not significant as more variables were added, and the SD of openness to experiences. The regression that included all the independent variables had a $R^2$ of 0.346 (Table 3, column 5).

Dividing the sample into the Gallup group subsamples produced too few respondents in the suffering group (n = 81) to run a reliable regression.

## Discussion

Considerable research has been done on the contribution of individual characteristics and the environment to self-reported life satisfaction (LS) over the past two decades [18–22]. Our results extended this to look at how the stability (and by proxy resilience and set-point) of an

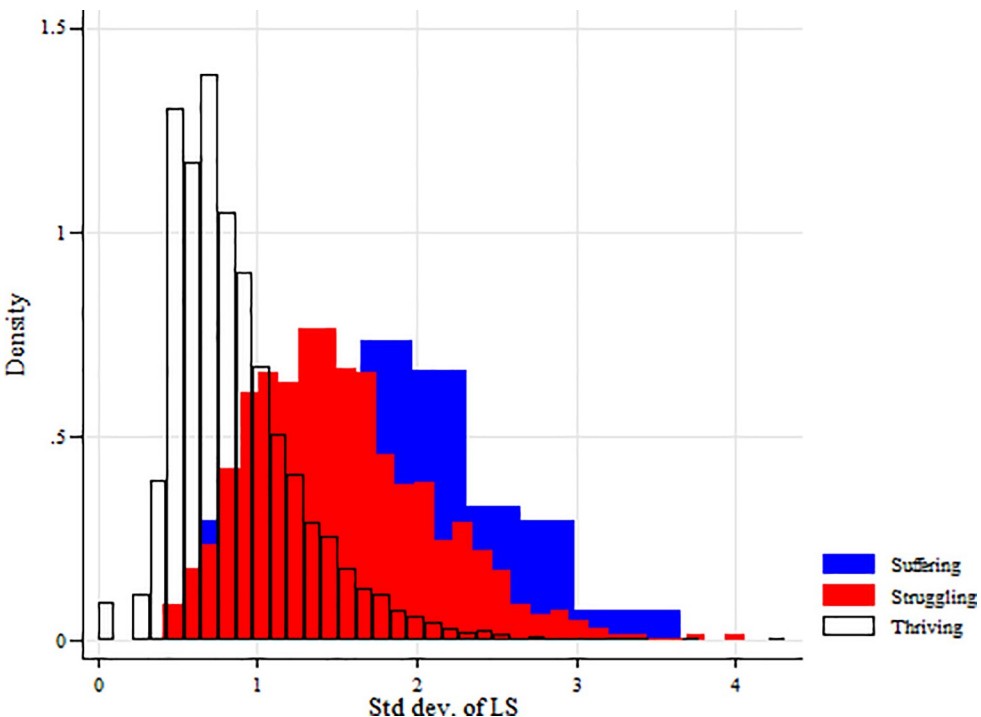

**Fig 3. The distribution of SD of LS for the three groups.** Individuals in the suffering range had an average SD of 1.96, those in the struggling range an average of 1.54, and those that were thriving an average SD of 0.87.

individual's LS depends on how satisfied they are with their lives, on average over a given period, and their personality traits. In this, and the next, sections we unpack some of these results and discuss their policy implications.

### Average life satisfaction

We found that an individual's resilience to life events is impacted by their average LS. Table 2 shows that average LS is significantly and negatively correlated with the standard deviation of LS (SD of LS). This means that these individuals are more susceptible to life events impacting their LS, both positively and negatively. If their LS can be impacted positively, opportunities exist for targeted policies to improve overall LS of communities that are struggling [44, 45]. Identifying factors that detract from those individuals' lives or communities, and making even minor interventions in addressing those issues, could have significant impacts on the LS of the entire Australian community.

We found that for those individuals that are suffering (LS = 0–4.4), LS and SD of LS can be seen to be much more variable, where shifts in both are significant in both the positive and negative directions. Both these trends decrease for individuals that are struggling (LS = 4.5–6.4) and decrease even more so for individuals that are thriving (LS = 6.5–10). Moreover, the relationship between average LS and SD of LS is robust with the inclusion of other covariates such as personality traits, income, and age.

This all implies that improving LS for those in the suffering and struggling ranges can improve not only their short term LS, but their resilience to future shocks. However, individuals in the lower ranges of LS may also be "trapped" and overcoming these traps may be more difficult than simply raising income. For example, personality is an important factor that may require a completely different approach (i.e. mindfulness training), Probably a more integrated

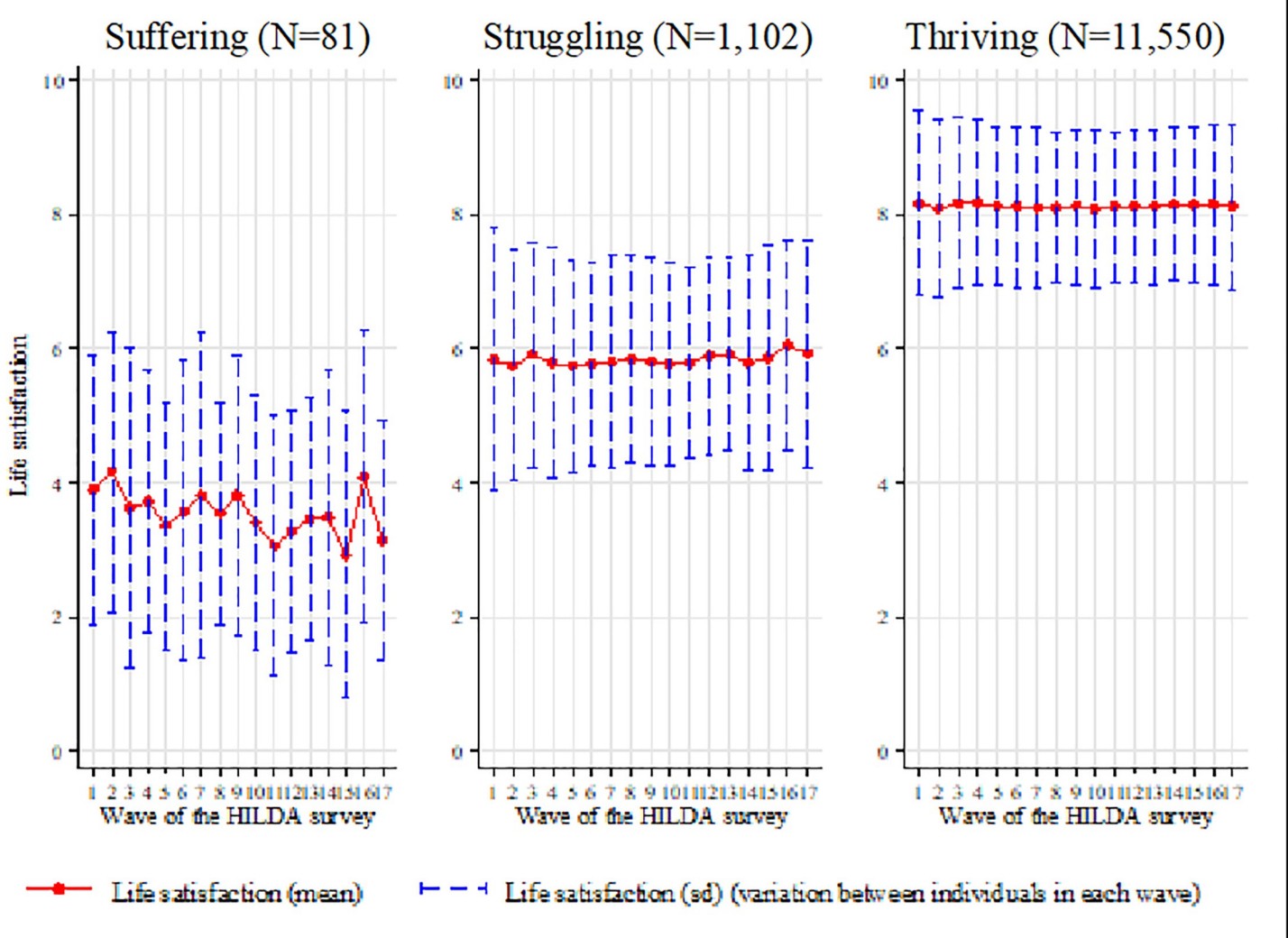

**Fig 4. Average and standard deviation of life satisfaction across individuals over the 17 waves of the HILDA survey divided into suffering (0–4.4), struggling (4.5–6.4) and thriving (6.5–10).**

approach to improving LS and resilience is needed. Our findings are a step in developing that more integrated approach.

## Personality

We found that personality is an important factor in determining how stable an individual's LS is. Looking at the Big Five personality traits (extroversion, agreeableness, conscientiousness, emotional stability, and openness to experience), on average Australians fall near the middle of each category, with averages between 4.19 and 5.39 (Table 2, Column 4). Most individuals' in this study show little personality change across the years in which the personality questions were asked: 2005, 2009, 2013, and 2017 (Table 2, Column 8). Previous research has found that over the course of an individual's life span, personalities may shift, especially when entering adulthood and seniorhood [35, 55, 56], but are in general fairly stable.

**Table 3. Multiple regressions between an individual's standard deviation in life satisfaction (all).**

| | (1) | (2) | (3) | (4) | (5) |
|---|---|---|---|---|---|
| Life satisfaction (mean) | -0.221*** | -0.226*** | -0.227*** | -0.228*** | -0.216*** |
| | (0.00422) | (0.00474) | (0.00476) | (0.00488) | (0.00478) |
| Extroversion (mean) | | 0.0157*** | 0.0140*** | 0.0128** | 0.0223*** |
| | | (0.00401) | (0.00403) | (0.00402) | (0.00394) |
| Agreeableness (mean) | | 0.0452*** | 0.0468*** | 0.0601*** | 0.0501*** |
| | | (0.00584) | (0.00582) | (0.00597) | (0.00579) |
| Conscientious (mean) | | -0.0130** | -0.0109* | 0.000449 | 0.00455 |
| | | (0.00465) | (0.00467) | (0.00468) | (0.00453) |
| Emotional stability (mean) | | -0.00872 | -0.00665 | 0.00298 | -0.0157** |
| | | (0.00491) | (0.00494) | (0.00503) | (0.00497) |
| Openness to experience (mean) | | -0.0359*** | -0.0363*** | -0.0166*** | -0.00998* |
| | | (0.00451) | (0.00449) | (0.00440) | (0.00427) |
| Extroversion (std.dev.) | | | | 0.131*** | 0.119*** |
| | | | | (0.0141) | (0.0138) |
| Agreeableness (std.dev.) | | | | 0.139*** | 0.104*** |
| | | | | (0.0140) | (0.0139) |
| Conscientious (std.dev.) | | | | 0.0660*** | 0.0579*** |
| | | | | (0.0141) | (0.0137) |
| Emotional stability (std.dev.) | | | | 0.0969*** | 0.0888*** |
| | | | | (0.0128) | (0.0126) |
| Openness to experience (std.dev.) | | | | 0.136*** | 0.122*** |
| | | | | (0.0146) | (0.0142) |
| Log income (mean) | | | | | -0.231*** |
| | | | | | (0.0110) |
| Log income (std.dev.) | | | | | 0.0804*** |
| | | | | | (0.00777) |
| Age (mean indiv.) | | | | | 0.00129*** |
| | | | | | (0.000265) |
| Controlled for waves responded | No | No | Yes | Yes | Yes |
| Observations | 12643 | 12316 | 12316 | 11337 | 11328 |
| Adjusted $R^2$ | 0.233 | 0.240 | 0.243 | 0.299 | 0.346 |

This table show the results of correlations between the standard deviations within each individual over time and five different sets of independent variables. Robust standard errors in parentheses.

* $p < .05$

** $p < .01$

*** $p < .001$.

When looking at correlations between SD of LS and the Big Five personality trait scores for the entire sample, we found that four of the personality traits were significant—all but conscientious. Extraversion and agreeableness were both negatively correlated with resilience while emotional stability and openness were positively correlated. We can only offer some possible reasons for these correlations.

We found that the more extraverted and agreeable an individual is, the higher the SD of LS, and lower the resilience to life events. This is interesting, since, in general, extraversion and agreeableness are positively correlated with LS itself.

This may be because introverts are less reliant on others for their LS. Extraverts are more sociable [57]. With greater sociability, extraverts have higher LS in general, but leave themselves open to experiencing more positive and negative events in their relation with others [58]. Individuals with higher agreeableness may be more prone to fluctuations in their LS as they are more willing to attempt adaptive coping mechanisms, depending on the type of life event occurring, with potentially more or less success [59]. If those coping mechanisms are not successful, they will experience greater fluctuations in their LS.

Emotional stability and openness to experiences were both significantly and positively correlated with resilience. This is in line with the generally positive relationship between higher emotional stability and openness with higher LS. This may be because individuals with higher emotional stability react less emotionally to negative events and are able to maintain a higher LS over time. Individuals that are more open to experiences may be able to adapt their responses to new life circumstances more efficiently allowing them to maintain greater stability on their LS [60].

While personality is thought to be relatively stable, there are emerging methods to change certain characteristics to improve LS (see below). Our results may help to better understand the implications of those methods.

## Set-point theory and resilience

Previous research has shown that some individuals experience a large decrease in LS after a major dramatic life event, while others seem to show little change in their LS [41]. We show that these differences are partially due to a higher average LS and personality traits that facilitate higher resilience. Previous work has also shown that personality traits alone don't completely moderate changes in LS to certain major events [32, 61]. However, combining those with average LS, household income, and age, we explained about 35% ($R^2$ = 0.346) of the stability in LS (Table 3, column 5).

Average LS alone explains approximately 23% ($R^2$ = 0.233) of this stability in LS (Table 3, column 1), adding just personality, increases this to 24% ($R^2$ = 0.240) (Table 3, column 2). This indicates that average LS has the largest impact on an individual's resilience. So, while many variables influence an individual's LS, it is also important to understand the factors that affect the stability of that LS, as we have attempted to do. Improving LS not only improves LS itself, but also the stability of that LS to life events.

## Policy implications

Approximately 30–40% of an individual's LS can be attributed to the characteristics of individuals, like personality, which are more difficult to change with policy interventions [18–22]. However, studies have shown that cognitive therapy, mindfulness, meditation, and other intentional influences can promote prosocial behaviour and increase overall wellbeing [62–64]. This is especially true in children, with educational interventions around emotional training in schools [65–67]. This implies that better and more meaningful education aimed not only at technical skills, but also social and mental coping skills, could improve LS and resilience later in life.

Understanding how an individual's personality allows them to interact with society can help to inform policies. For example, ensuring access to social networks and encouraging social interaction could raise the LS for suffering introverts and make them more resilient.

The majority (60–70%) of an individual's LS can be attributed to environmental conditions, which a government can influence through policies [18–22]. The role of government is to ensure that those environmental conditions benefit the population, to the extent possible [68].

Appropriate policies can significantly impact LS of a population. On the other hand, measuring LS can help determine the effectiveness of targeted policies and their overall impact on a community's wellbeing [69].

Targeted policies, implemented at the appropriate scales, can improve individuals' average LS scores for portions of the populations most in need. These policies can also make individuals more resilient to other changing life events. Next, we propose some policies that have been shown to improve LS.

## Inequality of life satisfaction

Identifying those individuals and communities that are the most as risk, is critical not only for those individuals, but for all individuals within the society. A high inequality of LS across the population has been shown to negatively impact the LS of everyone and be costly to society [44, 45, 50]. Seeing a neighbour struggling while you are thriving, or vice versa, will decrease your overall satisfaction. Identifying those individuals that are suffering and struggling, or are the most at risk, is a critical first step to ensuring that not only those people are taken care of, but that the entire population is healthy.

We found that the percentage of people suffering decreased from 3.32 to 2.68% and those struggling decreased from 11.29 to 9.77% over the past 17 years, while those thriving increased from 85.39 to 87.64% over this period. This implies that in Australia, inequality of LS has decreased slightly over time [44]. The fact that the inequality of LS has decreased in Australia shows that previously implemented policies have been successful to some extent in improving the LS for many.

Addressing inequality of LS can increase the LS of the entire population [44]. By identifying neighbourhoods and communities that have the lowest LS, and hence are the least resilient to changing circumstances, steps can be taken to address specific conditions. This provides an opportunity to minimize the number of individuals that are suffering through policy interventions at appropriate policy scales to ensure policy effectiveness. There will also be decreasing marginal returns to LS improvement, so it may be better to focus attention on improving the LS of those with low scores. National or state policies ignore spatial heterogeneity of a population and may hide marginal responses to a policy [44, 45].

## Social capital

Social capital has also been shown to have a significantly positive impact on LS [70–72]. Government policies around urban planning can affect the quality and quantity of social interactions through neighbourhood design that increases common greenspaces for individuals to socialize [73–75]. One of the greatest contributors to LS is exercise [45, 76]. Through proper urban planning, running paths and exercise areas can facilitate and encourage healthier mental and physical health. Community centres can increase the sense of community and provide local populations with meeting places and structured group activities [77]. Facilitating the development of community and co-housing projects can also provide social capital through facilitation of multi-generational interaction [78, 79].

## Job security

Although many individuals do return to a set-point in LS, one of the most consistent life events that is an exception to this theory is long-term unemployment. Although income insecurity is a part of this long-term reduction in LS, a far more substantial cause is the loss of social status, self-esteem, and professional networks [15, 16, 23]. Countries that have provided strong social protection or security net through programs with generous benefits for individuals that lose

their jobs and facilitate the return to work have been shown to have higher levels of LS [15]. This also increases LS of those employed by reducing job insecurity and fear of unemployment [80]. In general, higher LS has been found to be strongly correlated with low unemployment, implying that well designed fiscal policies can strongly impact LS [81–83]. Overall, austerity policies have led to a decrease in wellbeing through deteriorating mental and physical health [84].

## GDP and life satisfaction

Focusing on increasing GDP alone has been shown to have little effect on LS, and wellbeing in developed countries. For example, since the 1980s, China has undergone a major transition towards capitalism. This transition opened economic markets and significantly increased the country's GDP. However, it also degraded the natural environment, increased income inequality, increased unemployment, and took away the social safety net that had existed in China [85]. Although, this transition increased LS of the overall Chinese population initially, a decrease in LS has been seen in recent years [85]. A similar trend can be seen when looking at other indicators for China, such as the Genuine Progress Indicator which incorporates the effects of income inequality and damages to social and natural capital [86].

## Conclusions

We have shown that set-point theory applies more to individuals whose LS is already high and whose personality traits (specifically emotional stability and openness) facilitate higher resilience. The policy implications of these results are important. If our goal is to improve the LS of the population, we can make more progress by focusing on those who are suffering with low LS rather than those who are already thriving. This can improve equity in LS and that, in itself, can improve everyone's subjective wellbeing. This has been happening in Australia over the last two decades [44]. These results are critical in designing policies aimed at improving people's lives.

We can also better recognize the relationships between personality characteristics and subjective wellbeing. This is important in two ways: (1) it allows us to recognize that the same life conditions will affect people with different personality differently, so we need to acknowledge this when evaluating the overall LS of a country or region; and (2) personality characteristics in individuals *can be changed*, especially in early childhood. Investment in better education (at all ages) around prosocial behaviour, mindfulness, and coping skills can improve emotional stability and openness and thus improve LS for individuals.

Much of current research is based on surveys of individual's overall satisfaction with their own individual lives [20, 50, 87, 88]. Individual's interactions with their friends, families, communities, and society at large are assumed to be incorporated in these assessments, but we could take a broader, more interactive approach. Prosocial behaviour is often facilitated by deliberative processes that allow individuals to interact with each other. If we wish to assess community (or national) wellbeing, should we not ask the community? This should be more than just the aggregate of individual assessments. It could be the result of community deliberation and discussion. In addition, there may be strong geographical differences in resilience. Do individuals in rural versus urban setting have stronger social connections and hence more resilience? Is isolation a factor in the resilience of wellbeing? We leave these important questions for future research.

## Supporting information

**S1 Table. Summary statistics, BALANCED sample.** This table summarizes the primary variables for our 3 subsamples and the whole sample. Columns 1–4 provide the average across

individuals ($\overline{LS}_i$) of the average LS within an individual over time ($LS_{it}$) [see Eq 1]. The *(parenthesis)* provide the standard deviation between the average LS across individuals. Columns 5–8 provide the average across individuals of the standard deviations within each individual over time ($\sigma_i^{LS}$) [see Eq 2]. The *(parenthesis)* provide the standard deviation among the standard deviations of each individual over time.
(PDF)

**S2 Table. Robustness of our models.** Comparison between results from main (unbalanced) sample and the strongly balanced sample.
(PDF)

## Acknowledgments

This research was partially funded by the Australian Government through the Australian Research Council on a Discovery Early Career Researcher Award (Project ID: DE150100494). We thank Elizabeth Rieger and two anonymous reviewers for providing helpful comments on earlier drafts.

This paper uses unit record data from the Household, Income and Labour Dynamics in Australia (HILDA) Survey. The HILDA Project was initiated and is funded by the Australian Government Department of Social Services (DSS), and is managed by the Melbourne Institute of Applied Economic and Social Research (Melbourne Institute). The findings and views reported in this paper, however, are those of the author and should not be attributed to either DSS or the Melbourne Institute.

## Author Contributions

**Conceptualization:** Ida Kubiszewski, Nabeeh Zakariyya, Robert Costanza, Diane Jarvis.

**Data curation:** Nabeeh Zakariyya.

**Formal analysis:** Nabeeh Zakariyya.

**Funding acquisition:** Ida Kubiszewski.

**Supervision:** Ida Kubiszewski.

**Writing – original draft:** Ida Kubiszewski.

**Writing – review & editing:** Ida Kubiszewski, Robert Costanza, Diane Jarvis.

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
