## [Decision Letter · Decision Letter 0]

6 Jul 2020

PONE-D-20-14279

Resilience of Self-Reported Life Satisfaction: A case study of who conforms to set-point theory in Australia

PLOS ONE

Dear Dr. Kubiszewski,

Thank you for submitting your manuscript to PLOS ONE. After careful consideration, we feel that it has merit but does not fully meet PLOS ONE’s publication criteria as it currently stands. Therefore, we invite you to submit a revised version of the manuscript that addresses the points raised during the review process.

We look forward to receiving your revised manuscript.

Kind regards,

Frantisek Sudzina

Academic Editor

PLOS ONE

"This research was partially funded by the Australian Government through the

Australian Research Council on a Discovery Early Career Researcher Award (Project

ID: DE150100494)."

Reviewers' comments:

Reviewer's Responses to Questions

**Comments to the Author**

1. Is the manuscript technically sound, and do the data support the conclusions?

Reviewer #1: Yes

Reviewer #2: Yes

2. Has the statistical analysis been performed appropriately and rigorously? 

Reviewer #1: Yes

Reviewer #2: I Don't Know

3. Have the authors made all data underlying the findings in their manuscript fully available?

Reviewer #1: Yes

Reviewer #2: Yes

4. Is the manuscript presented in an intelligible fashion and written in standard English?

Reviewer #1: Yes

Reviewer #2: Yes

5. Review Comments to the Author

Reviewer #1: Overall comments

This is a really interesting paper exploring how self reported life satisfaction changes over time and how it is linked to personality traits. The introduction nicely sets out the wider context for this paper. However, the discussion seems to repeat a lot of the results without setting them into the wider context as well. The policy implications section is also but vague in places, there is the promise that specific policies will be discussed and recommended, but this seems to be lacking – in some cases these more specific policies are explained further in later sections, this could be better signposted in the earlier text. I'm surprised that there aren't any recommendations for further research included, survey approaches do have limitations, and it seems that the conclusions from this paper could be better supported with some additional qualitative research. The conclusion doesn’t quite manage to bring the paper together as it only briefly touches on the reduction in LS inequality, what about the findings on personality traits? What’s the bigger picture conclusion about those findings? I think the paper needs a few tweaks in the discussion and conclusion to be a really useful contribution. The conclusion mentions The paper also needs some proof reading to check that all acronyms are being used correctly and that the sentences are clear.

Whilst I understand the need for use of acronyms in scientific papers, I think in this paper it reduces the readability of some sections – I would recommend reducing the number to improve flow, for example by writing subjective well being out in full

Introduction

Overall this sets out the wider literature and context of the paper really well. But it would be useful to include a one sentence aim of what the paper sets out to do, what question are you answering?

Methods

The LS grouping into suffering, struggling and thriving is briefly mentioned on L180 but no context is given as to where these groups come from. In the results section, the authors state that the groups are derived from previous Gallup work – it would be clearer to explain this when the groupings are first mentioned in the methods section

Results

L229 – issue with referencing ‘error! Reference source not found’

Table 1 – It would be helpful to mention in the text (not just the table caption) where this data is from, and explain the differences compared to the HILDA findings – currently this seems a bit of a random add-on

Discussion

Text seems to switch between using LS or life satisfaction in the first paragraph

The first few sections seem to be repeating the results, it needs a bit more interpretation

P12 L4-7 – unclear sentence, individuals are more willing to attempt both more and less adaptive coping mechanisms – seems to contradict itself

The policy implications section is a bit vague in places e.g p13 L14-15, what sort of policies might have been successful in improving LS?

Conclusion

Again this mentions that Australia has been successful at reducing life satisfaction inequalities, but the paper did not manage to convey exactly how this might have been achieved

Reviewer #2: 1. Is the manuscript technically sound, and do the data support the conclusions?

Response: YES

Thank you for this clearly written, timely and interesting paper.

Some further questions for consideration, that might help strengthen the paper are included below:

A) I’m presuming the cut-off points for categorizing LS as suffering, struggling, or thriving were drawn from similar analyses used in the original survey and in the comparative country surveys (shown in the table) – it would be good to just state that in the methods to be clear.

B) In general, SD of Life Satisfaction (LS) as an indicator of resilience makes sense and the authors indicate that it is a neutral term, with the width of variation being the significant indicator. However, in the case of LS, it seems that the directionality of the variation would be of interest. For example, if in general SD was wider because it went “up” more rather than “down” it would seem to indicate a better outcome than if the SD was wide largely because it was trending down. So, if someone with a low set point had a wider SD (than someone with a higher set point), but it was wider because it was moving upwards, that would seem to indicate greater resilience in the face of traumatic events. This may reflect my own incomplete understanding of how SD was used however, and if so, the authors can disregard this comment.

C) The conclusion that “set point theory applies more to those individuals whose LS is already high and whose personality traits foster higher resilience” (i.e. those who are already happier, and have conducive personality traits, tend to be more resilient). seems supported by this research. The discussion on policy implications (addressing inequality in life satisfaction, social capital etc) are relevant and important). Addressing inequality of LS and policy implications is particularly relevant given recent health, social and economic crises (including covid 19’s unequal impacts, and heightened awareness of longstanding racial inequalities, e.g. BLM ). Is there scope to explore other possible policy implications? For example, the conclusion re: set point theory raises the question of whether early adverse life events or traumas may be important in shaping that set point – and therefore whether early life interventions (childhood, adolescence?) might be an important prevention / intervention point to elevate that set point later in life.

• It would be interesting to note whether there is any prior research or literature to support this

• It would be helpful to know whether there were any age-related effect in this research data

• Similarly to comment on whether there were any differences observed between men and women

D) The section on Personality is interesting, although the fact that the directionality of some of the associations with the big 5 traits are positively associated (and other negatively associated) could be seen as slightly counter-intuitive or confusing: (p.11)

• E.g. More extraversion and agreeableness = lower resilience (could be counter-intuitive if in extraversion and agreeableness are positively associated with higher mean LS)

• More emotional stability and more openness to experience = higher resilience (this is as one would expect)

If the big 5 are each positively associated with mean LS, one might presume that they would each also be positively associated with more resilient LS (smaller SD) as well. The authors describe how the directions of the associations they found could be explained in the Discussion with some references to supporting papers. It might have been interesting to know if the authors had any hypotheses regarding the directionality of these associations prior to the analysis. It might also benefit the discussion to cite whether there are any papers that support the opposite direction. (e.g. extraversion might be expected to be associated with higher resilience because individuals have a larger social network from whom to seek help during a crisis and are more likely to ask for help, resulting in more resilience)

(E) The authors note that 30-40% of LS can be attributed to individual traits such as personality which probably cannot be changed by policy interventions. However, to my understanding there is a growing literature indicating how educational interventions (social-emotional learning in schools, mindfulness and meditation training programs) can impact on neuroplasticity and other potential mechanisms for increasing wellbeing and resilience. It might be worth referencing some of this literature, as the implications for policy could be important.

For example, some research relating to this area:

https://www.nature.com/articles/nn.3093?page=10

https://pubmed.ncbi.nlm.nih.gov/24746260/

https://pubmed.ncbi.nlm.nih.gov/24746260/

https://www.researchgate.net/publication/327969562_Book_Review_-_Altered_Traits_Science_Reveals_how_meditation_changes_your_mind_brain_and_body

*2. Has the statistical analysis been performed appropriately and rigorously?

Response: I don’t know

In general, analysis methods seem sounds, however I do not have the statistical expertise to be able to comment fully on this aspect.

*3. Have the authors made all data underlying the findings in their manuscript fully available?

Yes – it appears that a link to the public repository for the data has been provided.

*4. Is the manuscript presented in an intelligible fashion and written in standard English?

Yes.

6. PLOS authors have the option to publish the peer review history of their article (what does this mean?). If published, this will include your full peer review and any attached files.

Reviewer #1: No

Reviewer #2: No

---

## [Editor Report · Decision Letter 1]

22 Jul 2020

Resilience of Self-Reported Life Satisfaction: A case study of who conforms to set-point theory in Australia

PONE-D-20-14279R1

Dear Dr. Kubiszewski,

We’re pleased to inform you that your manuscript has been judged scientifically suitable for publication and will be formally accepted for publication once it meets all outstanding technical requirements.

Kind regards,

Frantisek Sudzina

Academic Editor

PLOS ONE

---

## [Editor Report · Acceptance letter]

28 Jul 2020

PONE-D-20-14279R1 

Resilience of Self-Reported Life Satisfaction: A case study of who conforms to set-point theory in Australia 

Dear Dr. Kubiszewski:

I'm pleased to inform you that your manuscript has been deemed suitable for publication in PLOS ONE. Congratulations! Your manuscript is now with our production department. 

Kind regards, 

on behalf of

Dr. Frantisek Sudzina 

Academic Editor

PLOS ONE